# Parameters estimation of gas capture through Mixed Matrix Membrane (MMM) with CFD

**Ali A. Abdulabbas[1]\*, Thamer J. Mohammed[2], Tahseen A. Al-Hattab[3], Mahdi Sh. Jaafar[4]**

**1** Department of Chemical Engineering and Petroleum Industries, Al-Amarah University College, Maysan, Iraq, **2** Chemical Engineering Department, University of Technology, Baghdad, Iraq, **3** Chemical Engineering Department, College of Engineering, University of Babylon, Iraq, **4** Department of Chemical Engineering and Petroleum Industries, College of Engineering, Al- Mustaqbal University, Hilla, Iraq

\* che.20.02@grad.uotechnology.edu.iq

## Abstract

Carbon dioxide ($CO_2$) capture is a crucial process to mitigate greenhouse gas emissions and reduce anthropogenic impact on climate change. The 3-D model is choosing to capture carbon dioxide from real natural gas (NG) using a mixed matrix membrane (MMM) consisting of polysulfone (PSF) with nanoparticles of covalent organic frameworks (CT-1). In this work, computational fluid dynamics (CFD) estimated the parameters of MMM for $CO_2$ gas separation. Fick's law is utilized of gas transport over a membrane module, whereas the Navier-Stokes equation describes the gas transport in both the feed and permeate domains of the permeation cell. This study involves the estimation of the membrane's properties, including its permeance and diffusion coefficient. The estimation of these parameters was performed by integrating an artificial neural network (ANN) developed in MATLAB R2021a with computational fluid dynamics simulations in COMSOL 6.1. The goal of the parameter prediction module is to minimize the sum of squared errors (SSE) between the experimental and simulated concentrations in the permeate region. For different gas pairs with operating limitations, the calculated parameters for the MMM predict its performance. Additionally, the results showed that operational variables such as concentration of $CO_2$ and feed pressure have a direct impact on gas permeation, although temperature did not show a clear effect. According to the findings, the CFD model demonstrates a deviation of less than 5% from experimental data for the MMM in gas separation.

## 1. Introduction

In recent years, there has been a marked growth in the demand for natural gas on a worldwide basis [1]. Natural gas is the main contributor to carbon dioxide emissions, which heightens environmental and climate change concerns [2,3]. Carbon capture and storage could be a viable option for lowering natural gas's $CO_2$ emissions. In the

**Data availability statement:** All relevant data are within the paper and its Supporting Information files.

**Funding:** The author(s) received no specific funding for this work.

**Competing interests:** The authors have declared that no competing interests exist.

field of separation techniques, membrane units are also considered a suitable option because they are environmentally friendly, have low operating and capital expenditures, and use very little energy [4,5].

Mixed matrix membranes (MMM) with different types of structures are good for separating gases because they have fillers that are porous and have different functions [6]. Porous structures and functional groups facilitate gas movement as well as gas dissolution-diffusion, which overcomes low penetration and achieves very effective gas separation [7].

The use of nanomaterials is among the different types of methods for improving the membrane [8]. Nanomaterials are synthesized and mixed in a polymer solution to modify the phase composition and formation mechanism, creating pass channels leading to high-performance mixed matrix membranes(MMM) [9,10]. The introduction of inorganic substances in the membrane has been mainly limited due to the insufficient compatibility between the inorganic particles, and polymeric phase resulting in a drop in separation efficiency. Therefore, making organic porous materials with the functional groups could effectively solve these problems related to nanoparticles [11,12]. For instance, Gao et al. [13] utilized SNW-1, a COF filler, to prepare SNW-1/polysulfone (PSF) MMMs, which showed improved $CO_2$ permeation due to enhanced gas diffusion and $CO_2$ sorption properties. Similarly, Biswal et al. [14] developed COF/polybenzimidazole (PBI) MMMs with high $CO_2/N_2$ and $CO_2/CH_4$ selectivity, demonstrating the potential of COFs to improve gas separation performance. Thankamony et al. [15] further advanced this field by incorporating porous organic frameworks (CTPP) into PEBAX membranes, resulting in enhanced $CO_2$ permeability and selectivity. Despite these advancements, traditional empirical and semi-empirical models often fail to capture the intricate interactions between the polymer matrix, fillers, and gas molecules, leading to inaccurate predictions of membrane performance.

Computational fluid dynamics (CFD) has emerged as a powerful tool for modeling and simulating gas separation processes in membranes. CFD allows for the detailed analysis of fluid flow, mass transfer, and heat transfer within membrane modules, providing insights into the effects of various operational parameters on membrane performance such, as evaporation, combustion, condensation, chemical reactions, and crystallisation [16]. Furthermore, these models often rely on simplified assumptions that may not accurately represent real-world conditions, limiting their predictive capabilities. Shoghl et al. [17] provided a mathematical model to explain the phenomenon of the passage of gases across a polymeric membrane. With CFD, they calculated the law of continuity and the permeability flow of gas molecules across the membrane. Using a solution-diffusion process, the suggested model for polysulfone describes the gas's ideal gas behavior. The process is isothermal, steady-state, a single-dimensional and non-equilibrium sorption. The validity of the presented models was verified by experimental data. Qadir et al. [18] established 3D CFD model for the purpose of examining gas separation. They employed the COMSOL Multiphysics software for analyzing the gas flow through a module including a flat sheet membrane. The estimated outcomes of the suggested model were consistent with values that had been earlier published. Abdulabbas et al. [19] assessed the efficiency of a

polysulfone (PSF) membrane by employing CFD. The study focused on four suggested operational and design variables. The computational fluid dynamics (CFD) model accurately forecasts the spatial distribution of both the concentration and velocity of the individual components. Fick's law represents the gas transport process over the membrane, while the Navier-Stokes equation drives the flow of gases on both the inlet and permeate sides of the permeation unit. They examined the effects of gas flow rate, temperature, pressure, and membrane module diameter on the $CO_2$ mole fraction. A study by Tahmasbi et al. [20] used CFD model to guess how well silica membranes would work at separating hydrogen, which could be used as a source of clean energy. Takab and Nakao [21] applied the CFD technique to model the process of hydrogen and carbon monoxide separation over ceramic membranes. In their study, they developed numerous mathematical models to simulate gas separation by membranes, each based on unique assumptions [22]. The work also analyzed the performance of the membrane and studied the effects of various parameters such as temperature, pressure, internal radius, and flow rate of gases on the molar percentage of $H_2$.

Moreover, the integration of artificial neural networks (ANNs), with CFD simulations represents a promising approach to enhance the accuracy and efficiency of membrane performance predictions. ANNs are great for improving MMM design and operation because they can find complex, non-linear links between input parameters and membrane performance [23,24]. However, the literature doesn't go into enough detail about how hybrid CFD-ANN models can be used to separate gases, especially for MMMs. This study seeks to address these research gaps by developing a hybrid CFD-ANN model to estimate the permeance and diffusion coefficients of $CO_2$ and $CH_4$ in a polysulfone (PSF) membrane embedded with COF nanoparticles. The research aims to provide a more accurate and efficient method for predicting membrane performance by combining the strengths of CFD simulations and ANN-based parameter estimation. By investigating the effects of key operational parameters, such as feed pressure, temperature, and $CO_2$ concentration, on the separation performance of MMMs, this work contributes to the development of advanced gas separation technologies.

The significance of this work lies in its potential to enhance the understanding of gas transport mechanisms in MMMs and to provide a reliable tool for optimizing membrane design and operation. The findings could have broad implications for the natural gas industry, particularly in the context of $CO_2$ capture and storage, where efficient and cost-effective separation technologies are urgently needed. Furthermore, the integration of CFD and ANN techniques represents a novel approach to membrane modeling, offering a pathway for future research in the field of gas separation and beyond.

## 2. Experiment

In our previous work, the CT-1 mixed matrix membrane was produced, and the permeability values of the components were examined through the laboratory system [23]. Fig 1 illustrates the experimental setup employed for conducting gas permeation measurements. Gases methane ($CH_4$), and carbon dioxide ($CO_2$), were bought from Missan Oil Company in Iraq with levels of purity $\geq 99.4\%$. Accordingly, to study of Ali A. Abdulabbas [23], the permeance values are determined by measuring them under various operating settings, including varied concentrations of $CO_2$, temperatures, and pressures in the binary gas state.

The following equations were used to calculate gas permeability in a steady-state setting:

$$\frac{P_{CO2}}{l} = \frac{p_p y_{CO2}}{AT(p_f x_{CO2} - p y_{CO2})} \frac{dV}{dt} \tag{1}$$

$$\frac{P_{CH4}}{l} = \frac{p y_{CH4}}{AT(p_f x_{CH4} - p_p y_{CH4})} \frac{dV}{dt} \tag{2}$$

In this context, $p_f$ and $p_p$ indicate the supply and permeate pressures, respectively. A indicates the active area in $cm^2$. A bubble flowmeter measures soap-film volumetric movement as dV in $cm^3 s^{-1}$, T indicates the operating temperature of the

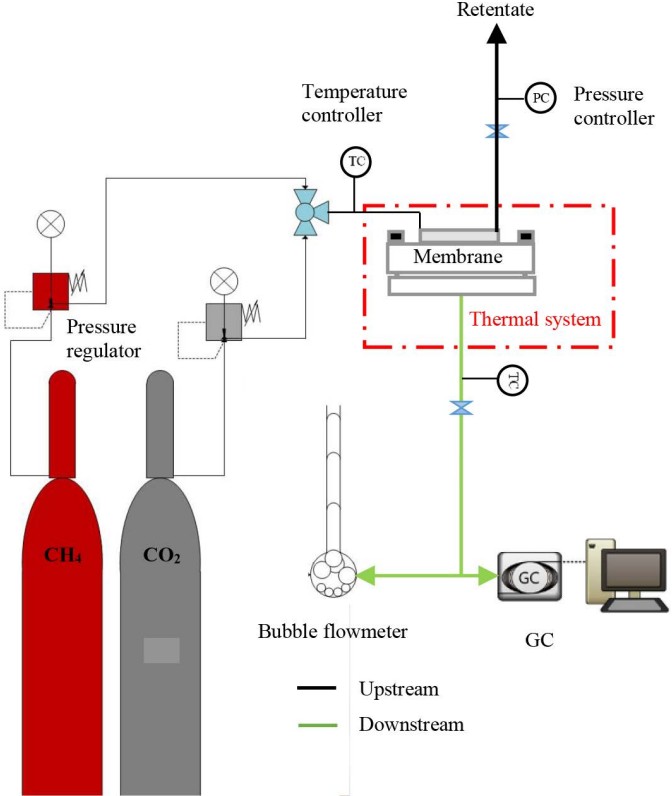

**Fig 1. Experimental set-up for gas permeation measurements.** The setup consists of a gas source, pressure control system, membrane module, and gas flow analysis unit.

feed (in K). The symbols x and y represent the mole fractions of gas on the feed side and the permeate side, respectively [24,25].

The Gas Permeation Units (GPUs), as described:

$$GPU = 1 \times 10^{-6} \frac{cm^3(STP)}{cm^2. \, s. \, cmHg}$$

The following calculation can be used to compute the selectivity of $CO_2$ relative to $CH_4$ gas:

$$\alpha_{\frac{CO2}{CH4}} = \frac{y_{CO2} \, y_{CH4}}{x_{CO2} \, x_{CH4}} \tag{3}$$

The parameters chosen were affected by the pressure and temperature requirements built into the membrane [19]. The $CO_2$ concentration was based on natural gas analysis from Maysan Oil Company fields in Iraq. Taguchi orthogonal array level 3 experiments were used to set the parameters. These were done by changing the gas input, which included the $CO_2$ concentration, temperature, and pressure, as shown in Table 1.

All of these parameters have predetermined ranges and discrete increments according to the experiments created in Minitab-19. All the experiments maintained a steady feed flow rate of 25 ml/min. In Table 2, the number of runs and the results from the experiments are displayed, including the percentage of carbon dioxide and methane in the reject.

   

**Table 1. The upper and lower limits of the different study settings.**

| No. | Pressure (bar) | Temperature (K) | $CO_2$ mol% |
|---|---|---|---|
| 1 | 2 | 293 | 3 |
| 2 | 3.5 | 313 | 9 |
| 3 | 5 | 333 | 15 |

**Table 2. The outcomes derived from the experiments accomplished.**

| Run | Inlet | | | | Permeate | |
|---|---|---|---|---|---|---|
| | Mole % | | Temperature(K) | Pressure (bar) | Mole % | |
| | $CH_4$ | $CO_2$ | | | $CO_2$ | $CH_4$ |
| 1 | 97 | 3 | 293 | 2 | 35.75 | 64.12 |
| 2 | 97 | 3 | 313 | 3.5 | 29.16 | 70.19 |
| 3 | 97 | 3 | 333 | 5 | 24.13 | 75.72 |
| 4 | 91 | 9 | 293 | 3.5 | 57.01 | 42.87 |
| 5 | 91 | 9 | 313 | 5 | 48.02 | 51.79 |
| 6 | 91 | 9 | 333 | 2 | 65.12 | 34.82 |
| 7 | 85 | 15 | 293 | 5 | 63.05 | 36.89 |
| 8 | 85 | 15 | 313 | 2 | 76.11 | 23.72 |
| 9 | 85 | 15 | 333 | 3.5 | 69.09 | 30.85 |

## 3. Model

### 3.1. Geometry and material balance

The membrane permeation was modeled by using computational fluid dynamics (CFD) while accounting for its real dimensions, which include an interior diameter of 40 mm, a length of 60 mm, and a total volume of 75398.22 mm³. Fig 2 illustrates the simplified design of the membrane module. The membrane was considered to separate the permeate and feed regions. The gas enters through the feed side, and the membrane selectively enables certain gas molecules to pass through based on specific passage mechanisms. Most of the gas was retained to gather impermeable particles. The following equation outlines the transport system's operation [26]:

$$j_i = P_m(p_{i,f} - p_{i,p}) \tag{4}$$

The variable $P_m$ refers the permeance of component i or j, $p_{i,f}$ and $p_{i,p}$ denote the partial pressure in the feed and permeate of gas. Lastly, $j_i$ as the molar flux.

The simulation and description model were established based on the following assumptions [27]:

1. Isothermal, ideal gas conditions, and steady-state are all necessary for the gas process to take place.

2. Fluid in three dimensions.

3. No chemical reactions are taking place on the membrane

4. Both supply and permeate gas flow are laminar.

5. The model is driven by pressure differences.

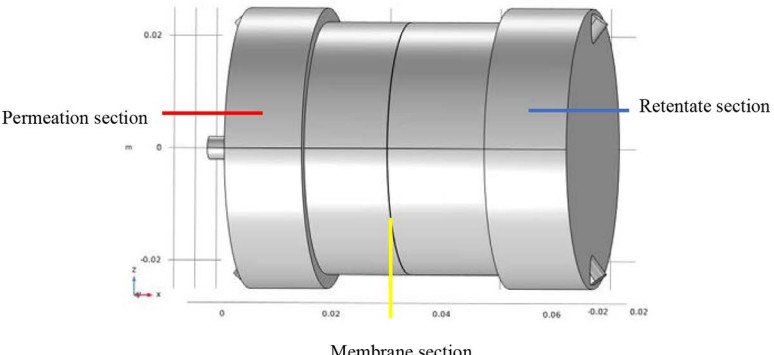

**Fig 2. Simplified design of the membrane module.** The model consists of a cylindrical membrane module with a feed and permeate region, allowing selective gas separation.

### 3.2. Governing equations

Every stage of the process is represented by one of three zones: feed, membrane, and permeate. The next part provides the guiding principles and mathematical equations for all scenarios. The governing equations employed for flow modelling are as follows [28,29]:

• Continuity equation:

$$\frac{\partial(\rho u)}{\partial t} + \nabla \cdot (\rho u) = 0 \tag{5}$$

• Momentum equation:

$$\frac{\partial(\rho u)}{\partial t} + \rho u(\nabla u) = -\nabla p + \nabla\left[\mu\left(\nabla u + (\nabla u)^T\right)\right] + F \tag{6}$$

where $p$ represents pressure, $\rho$ represents density, μ represents dynamic viscosity, $F$ *represents a body force*, and represents each of the three velocity components.

• Mass equations:

$$\rho\frac{\partial \omega_i}{\partial t} + \nabla \omega_i u = \nabla(D_{ij}\nabla\omega_i) \tag{7}$$

The two variables $D_{ij}$, and $\omega_i$ show the diffusion coefficient(i in j) and the mass fraction i, respectively. Equations (5)–(7) can be expressed as follows [30]:

$$\left(\frac{\partial\rho}{\partial t} + \frac{\partial(\rho u_x)}{\partial x} + \frac{\partial(\rho u_y)}{\partial y} + \frac{\partial(\rho u_z)}{\partial z}\right) = 0 \tag{8}$$

$$\left(\frac{\partial(\rho u_x)}{\partial t} + u_x\frac{\partial(\rho u_x)}{\partial x} + u_y\frac{\partial(\rho u_x)}{\partial y} + u_z\frac{\partial(\rho u_x)}{\partial z}\right) = -\frac{\partial P}{\partial x} + \frac{\partial}{\partial x}\left(\mu\frac{\partial u_x}{\partial x}\right) + \frac{\partial}{\partial y}\left(\mu\frac{\partial u_x}{\partial y}\right) + \frac{\partial}{\partial z}\left(\mu\frac{\partial u_x}{\partial z}\right) \tag{9}$$

$$\left(\frac{\partial(\rho u_y)}{\partial t} + u_x\frac{\partial(\rho u_y)}{\partial x} + u_y\frac{\partial(\rho u_y)}{\partial y} + u_z\frac{\partial(\rho u_y)}{\partial z}\right) = -\frac{\partial P}{\partial y} + \frac{\partial}{\partial x}\left(\mu\frac{\partial u_y}{\partial x}\right) + \frac{\partial}{\partial y}\left(\mu\frac{\partial u_y}{\partial y}\right) + \frac{\partial}{\partial z}\left(\mu\frac{\partial u_y}{\partial z}\right) \tag{10}$$

$$\left( \frac{\partial(\rho u_z)}{\partial t} + u_x \frac{\partial(\rho u_z)}{\partial x} + u_y \frac{\partial(\rho u_z)}{\partial y} + u_z \frac{\partial(\rho u_z)}{\partial z} \right) = -\frac{\partial P}{\partial z} + \frac{\partial}{\partial x}\left(\mu \frac{\partial u_z}{\partial x}\right) + \frac{\partial}{\partial y}\left(\mu \frac{\partial u_z}{\partial y}\right) + \frac{\partial}{\partial z}\left(\mu \frac{\partial u_z}{\partial z}\right)$$
(11)

$$\frac{\partial \omega_i}{\partial t} + u_x \frac{\partial \omega_i}{\partial x} + u_y \frac{\partial \omega_i}{\partial y} + u_z \frac{\partial \omega_i}{\partial z} = D_{ij}\left[ \frac{\partial^2 \omega_i}{\partial x^2} + \frac{\partial^2 \omega_i}{\partial y^2} + \frac{\partial^2 \omega_i}{\partial z^2} \right]$$
(12)

The axisymmetric of CFD model, as shown in Fig 3, occupies the 3D domain. Equations for controlling the feed and permeate sides of the CFD model are shown in Table 3.

### 3.3. Thermophysical properties

There are a number of correlations that are used to evaluate the binary gas mixture [28,31,32]:
Density,

$$\rho = \frac{pM}{RT}$$
(13)

Viscosity,

$$\mu = \sum_{i=1}^{n} \frac{\mu_i}{1 + \frac{1}{x_i} \sum_{j=1, j\neq i}^{n} x_j \phi_{ij}}$$
(14)

$$\mu_i = 2.669 \times 10^{-6} \frac{(M_i T \times 10^3)^{1/2}}{\Omega_D \sigma_i^2}$$
(15)

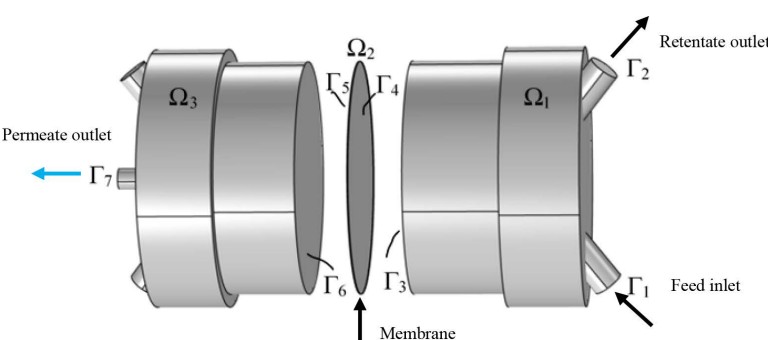

**Fig 3. Schematic diagram of the 3-D membrane model.** The figure presents a structured schematic of the simulated membrane module.

**Table 3. Boundary conditions for governing equations.**

| Domain | Position | Momentum and continuity | Mass transfer |
|---|---|---|---|
| $\Omega_1$ | $\Gamma_1$ | $u_{inlet} = u_0$ | $\omega_{i,in} = \omega_{0,i}$ |
| | $\Gamma_2$ | $p = p_f$ | $-n \cdot \rho D_i \nabla \omega_i = 0$ |
| | $\Gamma_3$ | $-j_{total} = -\sum j_i M_i$ | $-n \cdot j_i = -P_m(p_{i,\Omega 1} - p_{i,\Omega 2})M_i$ |
| $\Omega_2$ | $\Gamma_4$ | | $-n \cdot j_i = P_m(p_{i,\Omega 1} - p_{i,\Omega 2})M_i$ |
| | $\Gamma_5$ | | $-n \cdot j_i = -P_m(p_{i,\Omega 2} - p_{i,\Omega 3})M_i$ |
| $\Omega_3$ | $\Gamma_6$ | $j_{total} = \sum j_i M_i$ | $-n \cdot j_i = P_m(p_{i,\Omega 2} - p_{i,\Omega 3})M_i$ |
| | $\Gamma_7$ | $p = p_p$ | $-n \cdot \rho D_i \nabla \omega_i = 0$ |

$$\phi_{ij} = \frac{\left(1 + \left(\frac{\mu_i}{\mu_j}\right)^{1/2} \left(\frac{M_j}{M_i}\right)^{1/4}\right)}{\left(\frac{4}{\sqrt{2}}\right)\left(1 + \frac{M_i}{M_j}\right)^{1/2}}$$

(16)

Diffusion coefficient,

$$D_{ij} = 1.881 \times 10^{-3} \frac{\sqrt{T^3 \left(\frac{1}{M_i} + \frac{1}{M_j}\right)}}{P\sigma_{ij}^2 \Omega_D}$$

(17)

In the above context, the variables R, T, $\phi_{ij}$, $x_i$, and $M_i$ represent the universal gas constant, temperature, binding factor, the molar fraction, and molecular mass of component i, respectively. As illustrated in the equation, $\sigma_{ij}$ is an interaction parameter for a gas mixture [33]:

$$\sigma_{ij} = \frac{\sigma_i + \sigma_j}{2}$$

(18)

In terms of diffusion collisions, the integral expression is $\Omega_D$ [33]:

$$\Omega_D = \frac{b_1}{(T^*)^{b_2}} + \frac{b_3}{\exp(b_4 T^*)} + \frac{b_5}{\exp(b_6 T^*)} + \frac{4.998 \cdot 10^{-40} \mu^4{}_{Di}}{k_b^2 T^* \sigma_i^6}$$

(19)

In the given equation, the variable $T^*$ is denoted by:

$$T^* = \frac{T K_b}{\varepsilon_i}$$

(20)

The following equation uses the Lennard-Jones parameter, also known as $\mathcal{E}_i$, to calculate thermal conductivity (k) [32]:

$$K = 0.5 \left(\sum_i x_i K_i + \frac{1}{\sum_i \frac{x_i}{K_i}}\right)$$

(21)

In general, one can compute the diffusion coefficient in the PSF membrane by employing equations that make use of the fractional free volume (FFV) and Doolittle relations [34].

$$D_i = A \exp\left(-\frac{\beta_i}{FFV}\right)$$

(22)

$$FFV = \frac{\nu - \nu_o}{\nu}$$

(23)

The quantities denoted as $\nu$, and $\nu_o$ represent the molecule-occupied volume and specific volume, respectively [35]. The variables A and B are detailed in Table 4.

**Table 4. The values of the parameters β and A [36].**

| Gas | A (m²s⁻¹) | $\beta$ |
|---|---|---|
| $CH_4$ | $5.24 \times 10^{-10}$ | 1.19 |
| $CO_2$ | $2.08 \times 10^{-9}$ | 1.09 |

## 3.4. Parameter Estimation

The permeance and diffusion coefficients of membranes are critical in evaluating membrane performance, particularly for industrial processes such as $CO_2$ removal from natural gas and hydrogen purification. These properties directly influence separation efficiency, energy consumption, and operational costs [37].

Computational Fluid Dynamics (CFD) simulation is a widely used technique for modeling gas separation in membrane processes [38,39]. CFD allows for the calculation of mass transfer and fluid flow under varying operational conditions, including changes in pressure, temperature, and gas composition. This study employs a hybrid modeling approach, combining CFD simulations conducted in COMSOL 6.1 with Artificial Neural Networks (ANN) developed in MATLAB R2021a, to estimate membrane properties, specifically the permeance and diffusion coefficients for $CO_2$ and $CH_4$ in Mixed Matrix Membranes (MMM).

The first stage involved the development of a Computational Fluid Dynamics (CFD) model in COMSOL 6.1 to simulate gas transport through the membrane. The input parameters included:

1- Operating conditions: Pressure, temperature, and gas composition.

2- Membrane structure properties: Porosity, thickness, and material properties.

3- Unit design factors: Module dimensions and flow configuration.

These parameters are detailed in Table 5. To account for variability and uncertainty in the system, the Monte Carlo method was employed. This statistical approach simulates gas separation events (permeance and diffusion) by generating randomly distributed values within specified ranges for key parameters:

- Permeance: Ranging from $1 \times 10^{-9}$ to $1 \times 10^{-5}$ s·mol/(kg·m).

- Diffusion coefficient: Ranging from $1 \times 10^{-7}$ to $1 \times 10^{-4}$ m²/s.

In the second stage, the dataset generated by the CFD simulations (via the Monte Carlo method) was used as input for an Artificial Neural Network (ANN) designed for predictive membrane modeling. The ANN was developed in MATLAB R2021a, chosen for its ease of design and effectiveness in handling experimental data in chemical flows [40].

The ANN architecture consisted of a two-layer back-propagation network with 20 neurons in the hidden layer. A tangent sigmoid activation function was applied to the hidden layer, while a linear transformation was used in the output layer to convolve the parameters. The training process was guided by the Levenberg-Marquardt algorithm, using a mini-batch size of 32 and a maximum of 100 epochs. The objective was to minimize the sum of squared errors (SSE) between the predicted and experimental results, as shown in Equation 24:

$$\text{Error} = \sum (X_O - X_{des})^2$$

(24)

**Table 5. System configuration and operational specifications for the CFD simulation.**

| Specifications | | |
|---|---|---|
| The dimensions of design | Length, mm | 60 |
| | Diameter, mm | 40 |
| Membrane (CT-1(0.8)/PSF) | Porosity % | 56.4 |
| | Pressure, bar | 2, 3.5, 5 |
| Operating conditions | Temperature, K | 293, 313, 380 |
| | $CO_2$ concentration, mol% | 3, 9, 15 |
| | Flowrate, ml/min | 25 |

where $X_O$ and $X_{des}$ are the model's output and the experimental data for each required output.

The ANN was trained to predict four target outputs: $CO_2$ and $CH_4$ permeance, as well as $CO_2$ and $CH_4$ diffusion coefficients. In the final stage, the ANN outputs were used as inputs into the COMSOL software to study membrane behavior under various operating conditions. This hybrid approach, combining CFD and ANN, provides a reliable and efficient method for estimating membrane properties, making it highly suitable for complex gas separation applications as shown in Fig 4.

### 3.6. Grid independency

A mesh sensitivity test was performed by varying the grid cell numbers of the fluid domain. Grid independence was tested for average $CO_2$ permeation exit at varied mesh sizes. Fig 5 demonstrates that $CO_2$ permeation is not significantly different at mesh sizes above 80916. Our study used a large number of pieces to establish grid independence for the simulation.

### 3.7. Model Validation

The model validation results, comparing the simulation outcomes of this study with those of earlier studies [20], are shown in Figs 6 and 7. A comparison between the colour map of the present work's velocity distribution of $H_2/CO/CO_2$ gas and that of Ref. [20]. The comparison demonstrates a strong concurrence between the current study and the prior paper. The consensus among all the results was excellent.

## 4. Results and Discussion

### 4.1 Simulation of Paramters

The volume and concentration of ($CH_4$ and $CO_2$) permeated experimentally using MMM are displayed in Table 6. The permeance and diffusion coefficient simulation results were determined using the developed model, as indicated in Table 7. The results show that the permeance of $CO_2$ is significantly higher than that of $CH_4$, which is consistent with previous studies on gas separation using mixed matrix membranes (MMMs) [6,7]. This is primarily due to the smaller kinetic diameter of $CO_2$ and its higher affinity for the membrane material, which facilitates faster diffusion through the membrane.

The results indicate that the permeance of $CO_2$ increases with higher feed pressure, which is consistent with the findings of Qadir et al. [18], who also observed that increased pressure enhances the driving force for gas permeation.

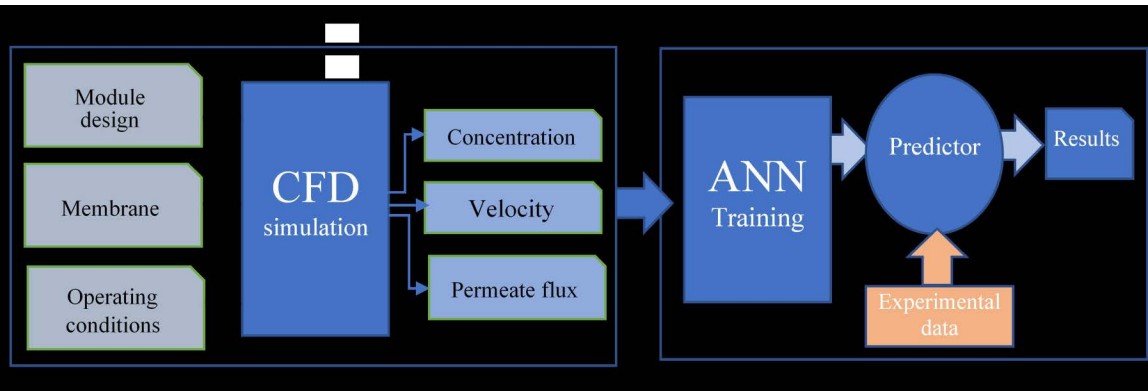

**Fig 4. ANN-CFD hybrid model integration.** The figure demonstrates the integration process between artificial neural networks (ANN) and computational fluid dynamics (CFD) for membrane performance prediction.

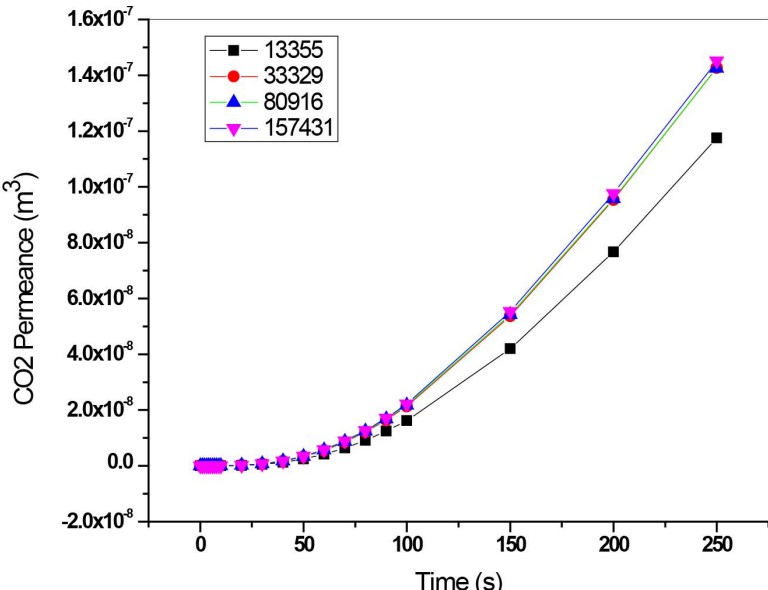

**Fig 5. Mesh sensitivity analysis for CO₂ permeation.** The figure presents the impact of different mesh sizes on CO₂ permeation.

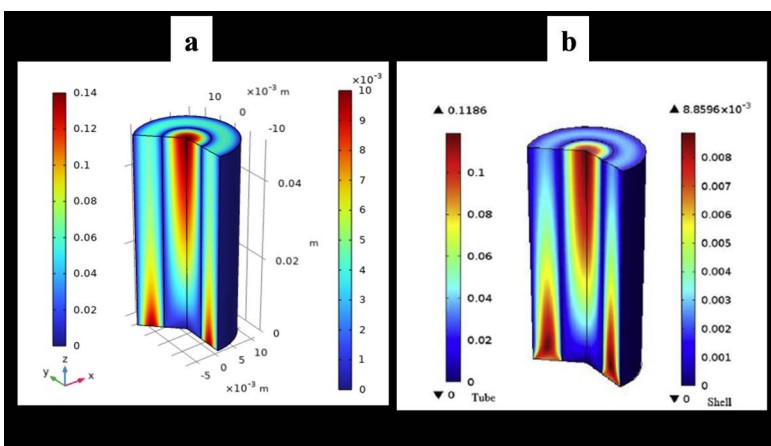

**Fig 6. Velocity distribution of H₂/CO/CO₂ gas.** The figure presents a comparison between the velocity distribution color map of the present study and that of Ref. [20].

However, the permeance of $CH_4$ remains relatively stable, which is likely due to its larger molecular size and lower diffusivity in the membrane material.

### 4.2 Simulation of gas permeation in mixed matrix membrane

Various operating settings were investigated using the model's mathematical equations and their associated boundary conditions for binary gas. The accuracy of the model was evaluated by comparing the experimental results with the model's predictions. For the purpose of applying the proposed model, four experiments (Run 1, 2, 5, and 8) from Table 2 were selected. In each of these experiments, the binary gas was introduced into the feed at different $CO_2$ concentrations,

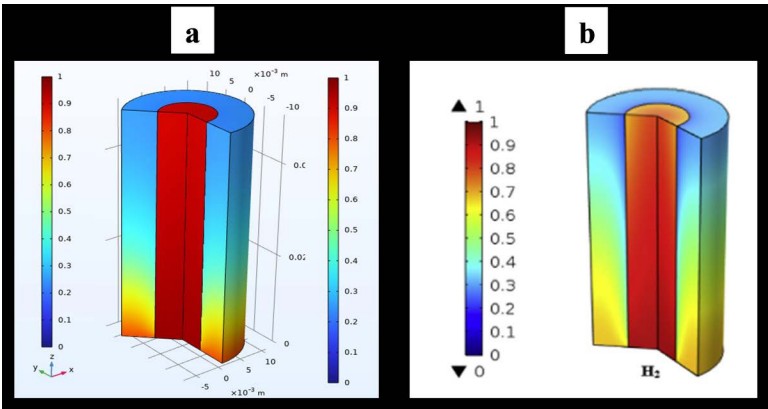

**Fig 7. Simulated molar fraction of hydrogen gas.** This figure illustrates the simulated molar fraction of hydrogen gas and its comparison with the numerical analysis results from Ref. [20], demonstrating strong agreement between the two studies.

**Table 6. The experimental data for the permeation of $CH_4$ and $CO_2$ through the membrane.**

| Run | Experimental data in membrane(MMM) | | | | | |
|---|---|---|---|---|---|---|
| | Inlet | | Permeate side | | | |
| | $CO_2$ mol % | $CH_4$ mol % | Time(s) | Volume(cm³) | $CO_2$ mol% | $CH_4$ mol% |
| 1 | 3 | 97 | 623 | 0.1875 | 35.75 | 64.12 |
| 2 | 3 | 97 | 411 | 0.1875 | 29.16 | 70.19 |
| 5 | 9 | 91 | 202 | 0.1875 | 48.02 | 51.97 |
| 8 | 15 | 85 | 251 | 0.1875 | 76.11 | 23.73 |

**Table 7. Simulation results for permeance and diffusion coefficient in membrane.**

| Run | Effective parameters | | | |
|---|---|---|---|---|
| | $CO_2$ gas | | $CH_4$ gas | |
| | Permeance P(s·mol/(kg·m)) | Diffusion coefficient D (m²/s) | Permeance P(s·mol/(kg·m)) | Diffusion coefficient D (m²/s) |
| 1 | $6.26 \times 10^{-10}$ | $2.11 \times 10^{-11}$ | $3.83 \times 10^{-11}$ | $1.21 \times 10^{-12}$ |
| 2 | $3.82 \times 10^{-10}$ | $7.41 \times 10^{-11}$ | $3.23 \times 10^{-11}$ | $3.85 \times 10^{-12}$ |
| 5 | $4.16 \times 10^{-10}$ | $9.96 \times 10^{-11}$ | $5.07 \times 10^{-11}$ | $7.08 \times 10^{-12}$ |
| 8 | $5.61 \times 10^{-10}$ | $4.05 \times 10^{-11}$ | $3.5 \times 10^{-11}$ | $1.83 \times 10^{-12}$ |

pressures, and temperatures. In order to verify the accuracy of the parameter estimate technique employed in this study, Table 8 displays a comparison between the experimental and anticipated values of the penetrated effluent of $CO_2$ gas. The upper and lower limits of the errors are 13.88% and 1.16%, respectively. The mean discrepancy between the reported result and the experimental data was calculated to be 6.89%. The cause can be attributed to the operational conditions and the content of the feed. This level of accuracy is comparable to previous studies, such as those by Tahmasbi et al. [20], who reported similar discrepancies in their CFD simulations of gas separation using silica membranes.

The results demonstrate that the model accurately predicts the permeation of $CO_2$ under various operating conditions, which is consistent with the findings of Abdulabbas et al. [19], who also reported good agreement between experimental and simulated data for $CO_2/CH_4$ separation using polysulfone membranes.

**Table 8. Evaluating the proposed model against experimental data.**

| Feed | | | | CO₂ (mol%) of permeate side | | Error (%) |
|---|---|---|---|---|---|---|
| CO₂ mol% | CH₄ mol% | Temperature(k) | Pressure(bar) | Experimental | Model | |
| 3 | 97 | 293 | 2 | 35.75 | 38.15 | 6.71 |
| 3 | 97 | 313 | 3.5 | 29.16 | 33.21 | 13.88 |
| 9 | 91 | 313 | 5 | 48.02 | 45.23 | 5.81 |
| 15 | 85 | 313 | 2 | 76.11 | 75.22 | 1.16 |

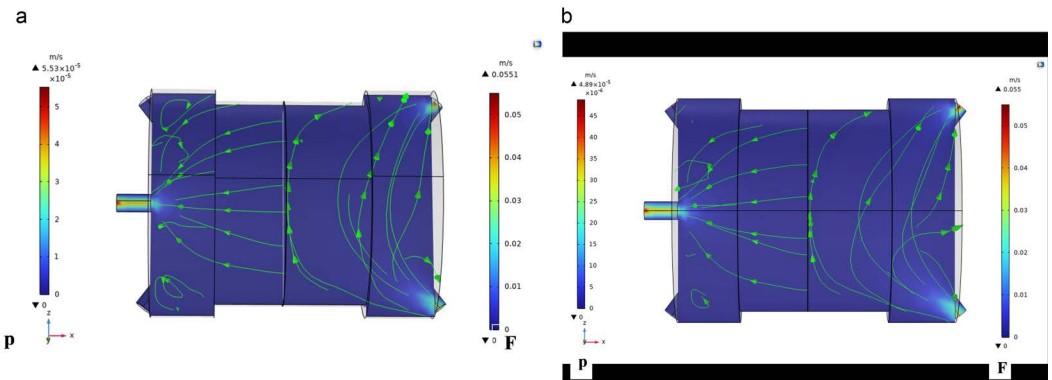

**Fig 8. Illustrates the velocity distribution at a flow rate of 25 ml/min and CO₂ concentration of 3% mol, with (a) T = 293K, p = 2 bar, and (b) at T = 313K, p = 3.5 bar.**

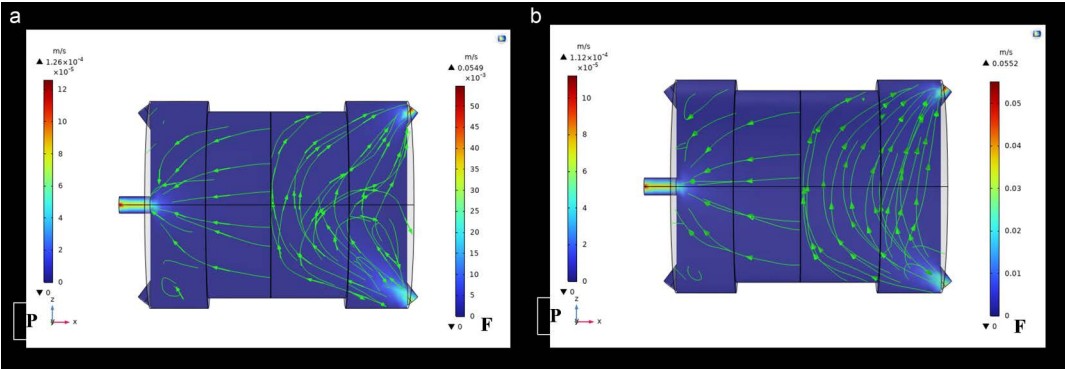

**Fig 9. Shows the velocity distribution at a fixed temperature of 313K and a flow rate of 25 ml/min, where (a) p = 5 bar, CO₂ = 9% mol, and (b) p = 2 bar, CO₂ = 15% mol.**

**4.2.1 Velocity distribution.** Under various temperature, pressure, and $CO_2$ concentration conditions, the CFD solved momentum calculations for the permeation system's feed and permeate sections. Navier-Stoke as equations were employed to find out the CFD model. The gas's velocity governs the convection-driven mass transfer on the feed side, as described by the continuity equation. On the other hand, the permeate side has a maximum value since the velocity increases gradually due to mass transfer across the membrane. Because the gas sweep was not present, the permeate-side velocity measurement was 0. Figs 8 and 9 present the velocity distribution color maps under different operating

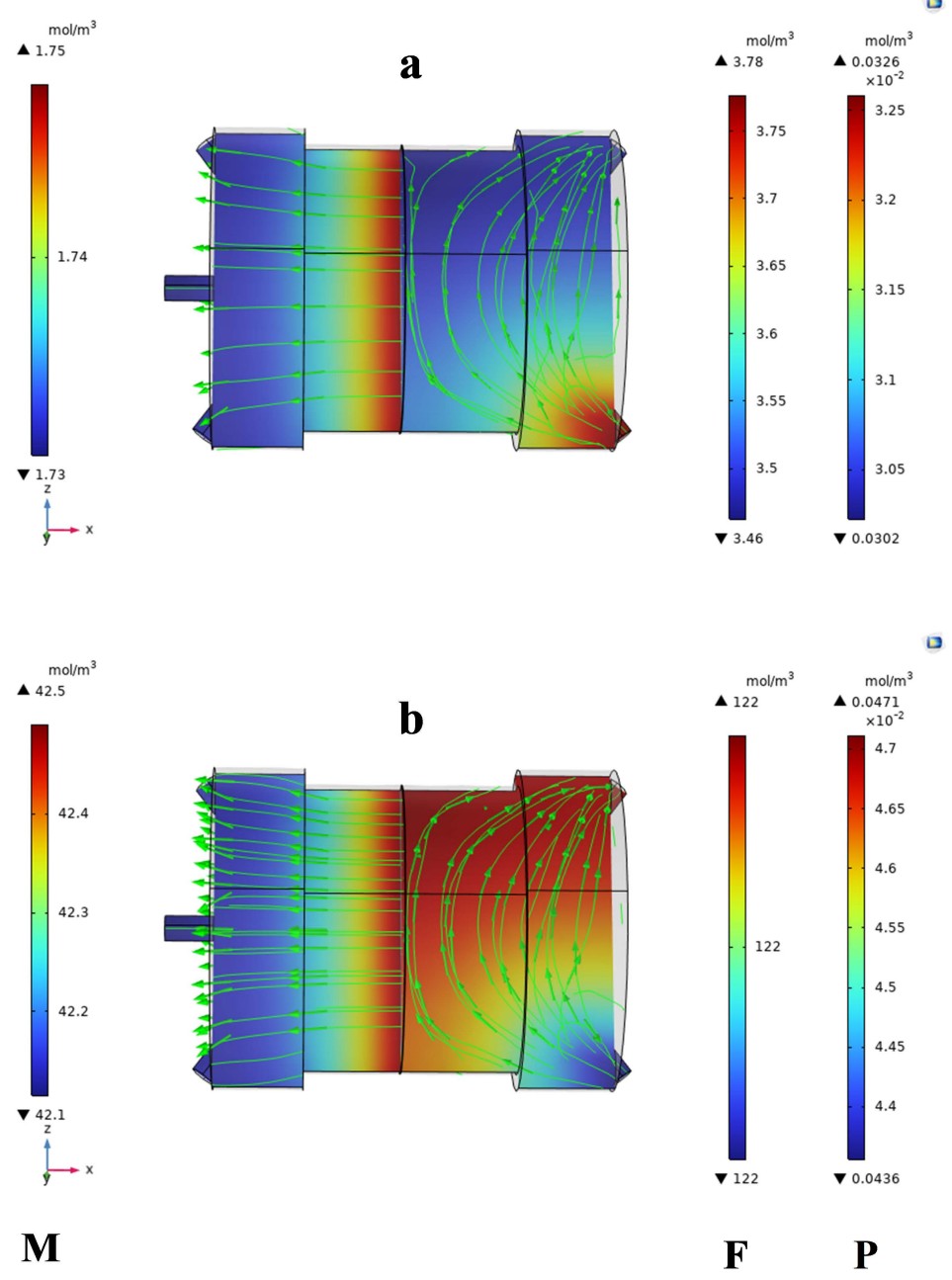

**Fig 10. Shows the concentration variations at 293K and 2 bar, with a flow rate of 25 ml/min and $CO_2 = 3\%$ mol.** (a) $CO_2$ concentration, (b) $CH_4$ concentration.

conditions. The results indicate that the velocity on the permeate side increases with higher feed pressure, which is consistent with the findings of Takaba and Nakao [18], who observed similar trends in their CFD simulations of gas separation using ceramic membranes. The increase in velocity is attributed to the higher driving force for gas permeation at elevated pressures [21].

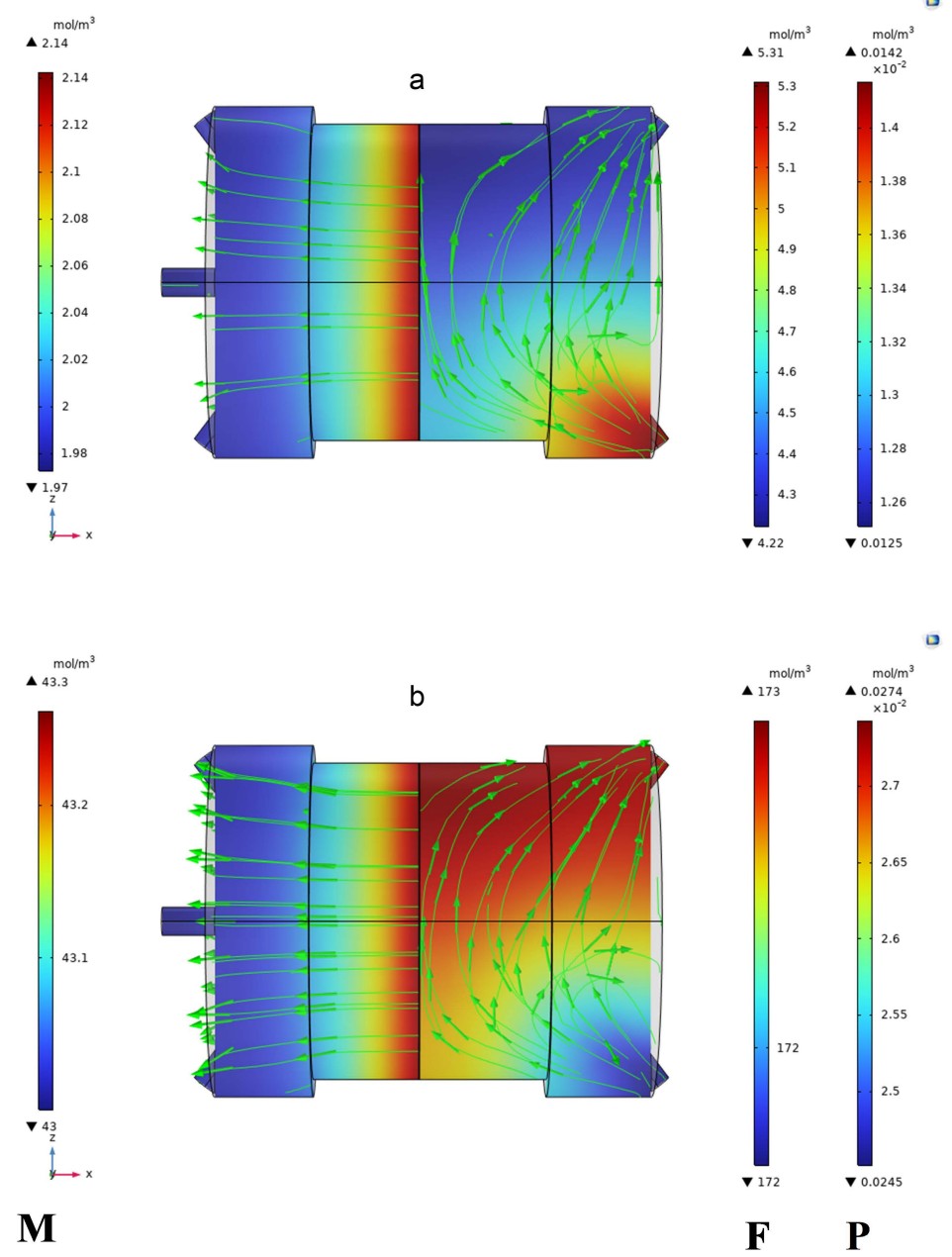

**Fig 11. Presents the concentration variations at 313K and 3.5 bar, with the same flow rate and CO₂ concentration.**

**4.2.2. Concentration distribution.** Typically, pilot plant or laboratory-scale testing employs the flat sheet membrane module. This study examined the separation of $CO_2$ and $CH_4$ using a flat-sheet membrane module. A simulation was conducted on a flat sheet membrane module to analyse the concentration variation on both the retentate and permeate sides. The feed gas was introduced into the membrane module, and the permeate was accumulated at the lower part of the module. A cross-flow model was used, incorporating specified boundary limitations.

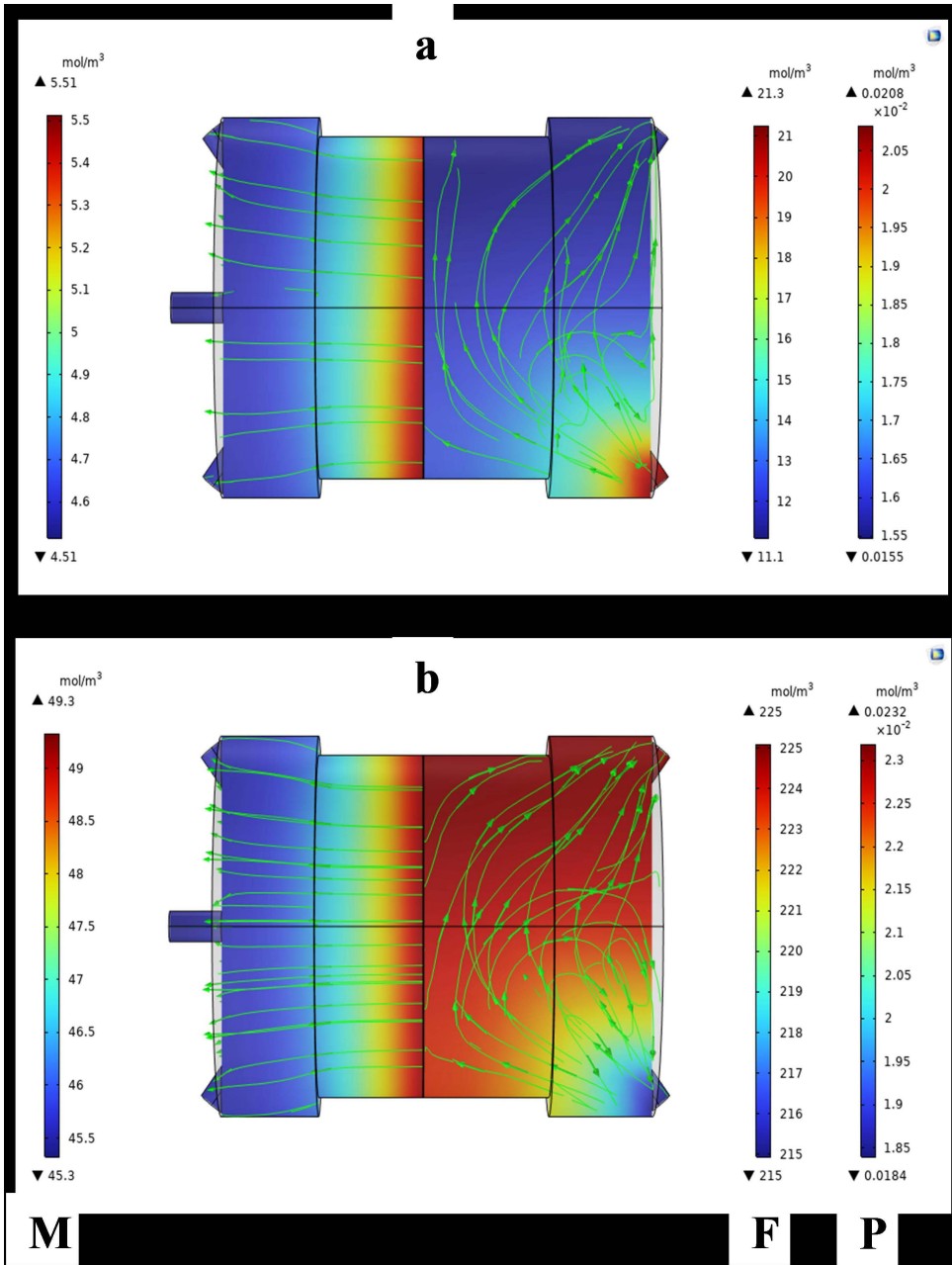

**Fig 12. Displays the concentration variations at 313K and 5 bar, with an increased CO₂ concentration of 9% mol.**

A computer model was performed to observe variations in concentration in the feed, membrane, and permeate sides. The formulae governing mass transfer in all three stages of the permeation unit were calculated under various operating limitations (containing the $CO_2$ concentration, pressure, and temperature as input variables) using CFD. In order to get the simulation results, add a gas consisting of carbon dioxide and methane as the feed on the right side. Prior to passing the membrane, the $CO_2$ gas content on the permeate side was zero.

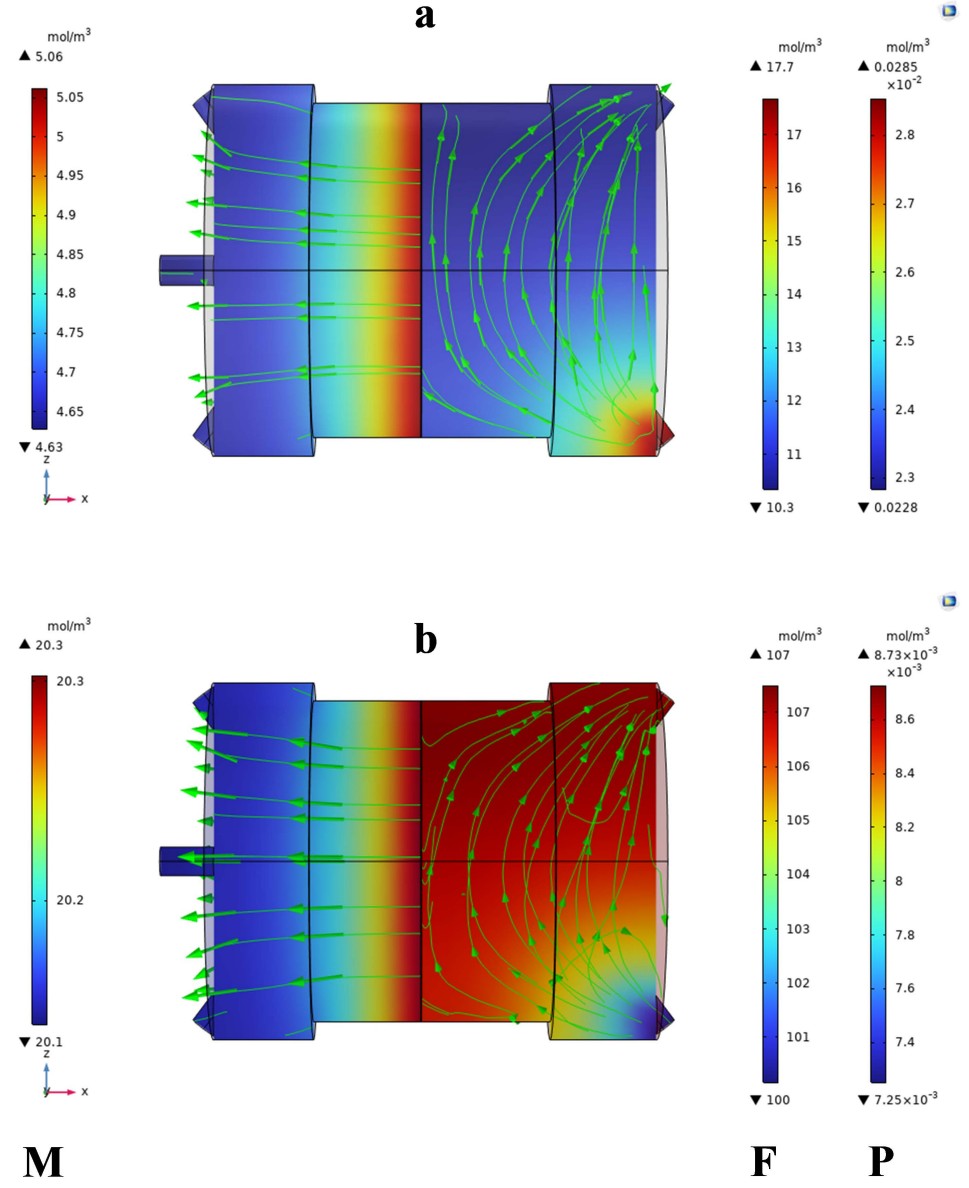

**M**  **F**  **P**

**Fig 13. Shows the concentration variations at 313K and 2 bar, with $CO_2$ = 15% mol.**

Figs 10–13Figs 10– illustrate the concentration variations of $CO_2$ and $CH_4$ under different input conditions. The results demonstrate that Carbon dioxide, although found in low concentrations, has a higher permeation rate than methane, which is consistent with the findings of Sun et al. [7], who reported similar behavior in their study of MOF-801 incorporated PEBA mixed-matrix membranes. The higher permeation rate of $CO_2$ is attributed to its smaller kinetic diameter and higher solubility in the membrane material.

The data clearly shows the gradient of $CO_2$ and $CH_4$ concentrations within the MMM module. The MMM module visually represented the concentration gradient using streamlines. The transition from high-to low-concentration areas is depicted

**Table 9. The value of the S/N ratio for permeance and selectivity.**

| Parameter | $(S/N)_{Selectivity}$ | | | $(S/N)_{Permeance}$ | | | $(S/N)_{Total}$ $= (S/N)_{Selectivity} + (S/N)_{Permeance}$ | | |
|---|---|---|---|---|---|---|---|---|---|
| | L1 | L2 | L3 | L1 | L2 | L3 | L1 | L2 | L3 |
| $CO_2$ (%) | 22.66 | 22.58 | 22.56 | 1.4944 | 1.9088 | 4.1039 | 24.1544 | 24.4888 | 26.6639 |
| Temperature | 22.56 | 22.48 | 22.75 | 1.1777 | 1.2978 | 1.0129 | 23.7377 | 23.7778 | 23.7629 |
| Pressure | 25.31 | 22.48 | 20.00 | 1.1523 | 0.6178 | -1.2924 | 26.4623 | 23.0978 | 18.7076 |

by the lines. Mass transfer occurs on both sides of the membrane through convection and diffusion, whereas gas transfer in membrane occurs only through diffusion.

### 4.3 Evaluation of gas separation in MMM

The statistical program (Minitab.19) was used to analyse the experimental outcomes presented in Table 2.

The objective of this examination was to study the impact of gas content, temperature, as well as pressure on the technique of separating carbon dioxide in the membrane. Table 9 compiles and presents the obtained signal-to-noise S/N ratios, with larger values indicating greater $CO_2$ permeance and selectivity. The investigation finds that using operational parameters such as 15 mol% $CO_2$, 313 K, and 2 bar results in the most effective separation performance.

At the lowest pressure, the $CO_2$ was highest. As the pressure increased, the $CO_2$ permeance within the module dropped. When gas flows across compressed membranes, the effective volume for the flow of gas decreases due to increased pressure, which explains the observed phenomena. In addition, a decrease in gas permeance is associated with a decrease in the mobility of polymer chains in high-pressure settings. The existing study also finds comparable results to previous studies [41].

The concentration of carbon dioxide impacts the ability of gases to pass through membranes. The concentration gradient induces mass transfer across the mixed matrix membrane (MMM). Dheyaa et al. [42] observed a strong correlation between the mole fraction in the permeate and the gas content in the feed.

Compared to other parameters, the impact of temperature on gas permeance is uncertain. In membrane-based $CO_2$ separation, temperature affects two opposing factors: solubility and diffusivity. As temperature increases, $CO_2$ solubility in the membrane decreases, while diffusivity increases. The decrease in solubility reduces the amount of $CO_2$ that can be absorbed, while the increase in diffusivity allows $CO_2$ to diffuse faster through the membrane. These competing effects often result in little or no significant change in the overall permeation rate, leading to a minimal impact of temperature on $CO_2$ separation performance in many cases [43].

## 5. Conclusions

In this study, the performance of permeable mixed matrix membranes (MMM) for capturing $CO_2$ was predicted using a 3D computational fluid dynamics (CFD) model. Methane and carbon dioxide were used in nine experiments to simulate composition natural gas. This study successfully developed a mathematical model to accurately simulate the mixed matrix membrane used for gas separation. Theoretical calculations were computed employing finite element method, and the outcomes for the mole fraction in the permeate assessed to experimental data to confirm their accuracy.

Membrane properties, permeability, and diffusion coefficient were estimated. When estimating these parameters, COMSOL 6.1 incorporates an artificial neural network (ANN) into the CFD simulation process. An experimental and theoretical investigation was conducted to study the separation of $CO_2$ using a mixed matrix membrane (MMM). In addition, the results demonstrated a clear correlation between the pressure and $CO_2$ concentration in the inflow stream and the penetration of gas. However, the temperature did not seem to have any noticeable impact. The findings demonstrate that

the computational fluid dynamics model is capable of precisely determining the parameters of the mixed matrix membrane and accurately forecasting its gas separation performance. The CFD model effectively predicts MMM performance in gas separation, highlighting the influence of operating and design factors. The model can predict membrane performance for different polymers and operating conditions and supports multiphysics modeling and hybrid simulation. However, high temperatures and pressures limit the usability of the model.

## Nomenclature

| $\rho$ | Density | kg m$^{-3}$ |
|---|---|---|
| u | Velocity | m s$^{-1}$ |
| $\mu$ | Viscosity | Pa s |
| T | Temperature | K |
| $u_x$, $u_y$, $u_z$ | Velocity x, y, z-axis | m s$^{-1}$ |
| $D_{ij}$ | Diffusion coefficient of i, j | m$^2$ s$^{-1}$ |
| $\mu_i$ | Viscosity of i | Pa s |
| $\sigma_{ij}$ | Collision diameter | m |
| $\Omega_D$ | Diffusion collision integral | – |
| $\varepsilon_i$ | Lennard Jones parameter | J |
| $K_i$ | Heat conductivity of i | W m$^{-1}$ K$^{-1}$ |
| K | Thermal conductivity | W m$^{-1}$ K$^{-1}$ |
| $j_i$ | Molar flux of i | mol m$^{-2}$ s$^{-1}$ |
| $j_{total}$ | Total mass flux | kg m$^{-2}$ s$^{-1}$ |
| $p_{i,f}$ | Feed-side partial pressure | Pa |
| $p_{i,p}$ | Permeate partial pressure | Pa |
| $p_{i,m}$ | Membrane partial pressure | Pa |
| $j_i$ | Mass flux of component i | kg m$^2$ s$^{-1}$ |
| $\omega_i$ | Mass fraction of i | – |
| $x_i$ | The mole fraction of component i | – |
| $M_i$ | Molar mass of i | kg mol$^{-1}$ |
| M | Molar mass | kg mol$^{-1}$ |
| $k_b$ | Constant of Boltzmann | J K$^{-1}$ |
| **Subscript** | | |
| 0 | Initial condition | |
| m | Membrane | |
| Exp | Experimental | |
| f | Feed | |
| p | Permeate | |

## Author contributions

**Data curation:** Thamer J. Mohammed.

**Formal analysis:** Ali A. Abdulabbas, Thamer J. Mohammed, Tahseen A. Al-Hattab, Mahdi Sh. Jaafar.

**Investigation:** Ali A. Abdulabbas, Thamer J. Mohammed, Tahseen A. Al-Hattab, Mahdi Sh. Jaafar.

**Methodology:** Ali A. Abdulabbas, Tahseen A. Al-Hattab, Mahdi Sh. Jaafar.

**Project administration:** Ali A. Abdulabbas, Tahseen A. Al-Hattab.

**Resources:** Ali A. Abdulabbas.

**Software:** Ali A. Abdulabbas, Thamer J. Mohammed, Tahseen A. Al-Hattab.

**Supervision:** Ali A. Abdulabbas, Thamer J. Mohammed, Tahseen A. Al-Hattab, Mahdi Sh. Jaafar.

**Validation:** Ali A. Abdulabbas, Thamer J. Mohammed, Tahseen A. Al-Hattab, Mahdi Sh. Jaafar.

**Visualization:** Tahseen A. Al-Hattab.

**Writing – original draft:** Ali A. Abdulabbas, Thamer J. Mohammed, Tahseen A. Al-Hattab.

**Writing – review & editing:** Ali A. Abdulabbas, Tahseen A. Al-Hattab.

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
