## [Decision Letter · Decision Letter 0]

26 Dec 2024

PONE-D-24-53065Parameters Estimation of Gas Capture Through Mixed Matrix Membrane (MMM) with CFDPLOS ONE

Dear Dr. Abdulabbas,

Thank you for submitting your manuscript to PLOS ONE. After careful consideration, we feel that it has merit but does not fully meet PLOS ONE’s publication criteria as it currently stands. Therefore, we invite you to submit a revised version of the manuscript that addresses the points raised during the review process.

We look forward to receiving your revised manuscript.

Kind regards,

Rizwan Nasir, PhD Chemical Engineering

Academic Editor

PLOS ONE

Additional Editor Comments:

The reviewers have suggested that the authors better highlight the novelty of their work. In the revised manuscript, the authors should focus on what makes their study unique and how it contributes to the field.

Reviewers' comments:

Reviewer's Responses to Questions

**Comments to the Author**

1. Is the manuscript technically sound, and do the data support the conclusions?

Reviewer #1: Yes

Reviewer #2: Yes

2. Has the statistical analysis been performed appropriately and rigorously? 

Reviewer #1: Yes

Reviewer #2: Yes

3. Have the authors made all data underlying the findings in their manuscript fully available?

Reviewer #1: Yes

Reviewer #2: No

4. Is the manuscript presented in an intelligible fashion and written in standard English?

Reviewer #1: Yes

Reviewer #2: Yes

5. Review Comments to the Author

Reviewer #1: This paper is well-written. Just need to add some information.

(1) The authors should expand the Introduction section to better identify research gaps in current literature and emphasize the significance of this work.

(2) Please specify the software versions used in this study (e.g., MATLAB Academic 2023 or other relevant software).

(3) Please provide more detailed information about the Artificial Neural Network (ANN) implementation, including:

• The dataset distribution (specify the number of samples used for training and validation)

• The network architecture (detail the number and configuration of layers)

(4) Enhance the Results and Discussion section by incorporating more references to relevant literature and comparing your findings with previous studies.

Reviewer #2: Summary of the Manuscript:

The manuscript titled "Parameters Estimation of Gas Capture Through Mixed Matrix Membrane (MMM) with CFD" investigates the potential of mixed matrix membranes (MMM) to capture carbon dioxide (CO₂) from natural gas using computational fluid dynamics (CFD). A 3D model was developed using COMSOL 6.1, combined with MATLAB's artificial neural network (ANN), to estimate critical parameters such as permeance and diffusion coefficients. The authors evaluated the impact of operating parameters such as feed pressure, temperature, and CO₂ concentration on the membrane’s performance. The work concludes that CFD modeling provides accurate predictions of gas separation performance for MMMs, while temperature exhibited minimal influence on separation efficiency.

Recommendation

The manuscript presents an important contribution to gas separation technologies, showcasing the integration of CFD and ANN in parameter estimation for MMMs. However, revisions are required to enhance the depth of analysis, methodological clarity, and overall presentation. I recommend major revisions to address the highlighted issues and improve the scientific rigor of the paper.

My comments are the following:

1. Expand on Novelty and Relevance: While the manuscript presents an innovative approach, the introduction could elaborate further on recent advancements in CO₂ capture using MMMs. Highlighting the novelty and the significance of the CFD-ANN integration would strengthen the impact of the study.

2. Enhance Methodological Clarity: Although the methods section mentions geometric models and computational tools, the description is somewhat brief. It is recommended to:

2.1 Provide more detail on the construction of the geometrical model used in the simulations.

2.2 Explain the rationale for selecting COMSOL 6.1 and MATLAB’s ANN for parameter estimation, particularly the advantages these tools offer over alternatives.

3. Validation and Comparative Analysis: The manuscript validates its model against one prior study, which limits confidence in the model's robustness. To address this:

3.1 Include comparisons with experimental or additional literature data to confirm the model’s reliability.

3.2 Provide a comparative analysis of MMM performance against other CO₂ capture methods (e.g., chemical absorption or alternative membranes). Tables or charts could enhance the clarity of these comparisons.

4. Analysis of Temperature Effects: The results section notes that temperature has minimal impact on separation performance, but the discussion is limited. A deeper analysis of how solubility and diffusivity counteract at varying temperatures would be beneficial.

5. Quantitative Metrics: The paper evaluates permeance and diffusion coefficients but lacks a broader context. Discuss how these metrics translate to industrial applicability, operational efficiency, and cost-effectiveness.

6. Expand Future Directions: The conclusion briefly mentions potential future work. To provide clearer guidance for subsequent studies, consider:

6.1 Identifying specific challenges for scaling up the MMM technology.

6.2 Suggesting experimental validations or additional computational studies for further refinement.

7. Ethics and Data Transparency: The manuscript does not mention ethical considerations or data availability. Even if ethics approvals are not required, this should be explicitly stated. Additionally, ensure all data complies with PLOS’s open data policy.

Minor Comments to follow to Enhance the Manuscript Overall Structure:

1- Correct typographical issues such as "Simlution" in Section 4.1 and ensure consistent use of terminology throughout the manuscript.

2- Improve figure captions to provide standalone clarity, including explanations for abbreviations and key observations.

3- Specify units for all parameters in equations to enhance reader comprehension.

4- Clarify ambiguous statements such as "precise parameter estimates" by providing accuracy thresholds or numerical ranges.

5- Include the rationale behind the chosen experimental conditions (e.g., pressure and CO₂ concentration) in the methodology.

6- Provide interpretations of figures directly in captions to help readers quickly grasp their significance.

6. PLOS authors have the option to publish the peer review history of their article (what does this mean? ). If published, this will include your full peer review and any attached files.

**Do you want your identity to be public for this peer review?** For information about this choice, including consent withdrawal, please see our Privacy Policy .

Reviewer #1: **Yes: ** Jingxian An

Reviewer #2: **Yes: ** Ms.Asma Alzarooni

---

## [Editor Report · Decision Letter 1]

12 Mar 2025

PONE-D-24-53065R1Parameters Estimation of Gas Capture Through Mixed Matrix Membrane (MMM) with CFDPLOS ONE

Dear Dr. Abdulabbas,

Thank you for submitting your manuscript to PLOS ONE. After careful consideration, we feel that it has merit but does not fully meet PLOS ONE’s publication criteria as it currently stands. Therefore, we invite you to submit a revised version of the manuscript that addresses the points raised during the review process.

In the abstract, authors should give a numerical value or percentage instead of precise parameter estimates.The second-to-last paragraph of the introduction section needs a well-cited reference.The formatting of the table needs to be checked. For example, in Table 9, there are no borders.There are spacing issues between text and reference numbers

We look forward to receiving your revised manuscript.

Kind regards,

Rizwan Nasir, PhD Chemical Engineering

Academic Editor

PLOS ONE

Journal Requirements:

Additional Editor Comments:

1. In the abstract, authors should give a numerical value or percentage instead of precise parameter estimates.

2. The second-to-last paragraph of the introduction section needs a well-cited reference.

3. The formatting of the table needs to be checked. For example, in Table 9, there are no borders.

4. There are spacing issues between text and reference numbers.

---

## [Author Response · Author response to Decision Letter 2]

14 Mar 2025

Response to Comments:

1- In the abstract, authors should give a numerical value or percentage instead of precise parameter estimates.

- Thank you for your comment. According to your comment, a percentage has been added.

2- The second-to-last paragraph of the introduction section needs a well-cited reference.

-Thank you for your comment. According to your comment, references were added.

3- The formatting of the table needs to be checked. For example, in Table 9, there are no borders.

Thank you for your comment. Borders have been added to the table.

4- There are spacing issues between text and reference numbers

Thanks for your comment. Based on your comment, we have made an edit.

Thank you again for reviewing our manuscript and for your valuable comments. We appreciate your suggestions and advice. We look forward to hearing from you soon.

Sincerely,

Ali A. Abdulabbas

Corresponding author

che.20.02@grad.uotechnology.edu.iq

---

## [Editor Report · Decision Letter 2]

17 Mar 2025

Parameters Estimation of Gas Capture Through Mixed Matrix Membrane (MMM) with CFD

PONE-D-24-53065R2

Dear Dr. Abdulabbas,

We’re pleased to inform you that your manuscript has been judged scientifically suitable for publication and will be formally accepted for publication once it meets all outstanding technical requirements.

Kind regards,

Rizwan Nasir, PhD Chemical Engineering

Academic Editor

PLOS ONE

Additional Editor Comments (optional):

The authors addressed all the raised comments satisfactorily. The manuscript can be accepted for publication after final an editorial check.
---

## [Editor Report · Acceptance letter]

PONE-D-24-53065R2

PLOS ONE

Dear Dr. Abdulabbas,

I'm pleased to inform you that your manuscript has been deemed suitable for publication in PLOS ONE. Congratulations! Your manuscript is now being handed over to our production team.

Kind regards,

on behalf of

Dr. Rizwan Nasir

Academic Editor

PLOS ONE